# Comparison of Changes in the Plumage and Body Condition, Egg Production, and Mortality of Different Non-Beak-Trimmed Pure Line Laying Hens during the Egg-Laying Period

**DOI:** 10.3390/ani11020500

**Published:** 2021-02-15

**Authors:** Gábor Milisits, Sándor Szász, Tamás Donkó, Zoltán Budai, Anita Almási, Olga Pőcze, Jolán Ujvári, Tamás Péter Farkas, Erik Garamvölgyi, Péter Horn, Zoltán Sütő

**Affiliations:** 1Institute of Animal Science, Kaposvár Campus, Szent István University, Guba Sándor Street 40, 7400 Kaposvár, Hungary; Szasz.Sandor@szie.hu (S.S.); Kustosne.Pocze.Olga@szie.hu (O.P.); Ujvari.Lajosne@szie.hu (J.U.); Farkas.Tamas.Peter@szie.hu (T.P.F.); Garamvolgyi.Erik@szie.hu (E.G.); Horn.Peter@szie.hu (P.H.); Suto.Zoltan@szie.hu (Z.S.); 2Medicopus Nonprofit Ltd., Guba Sándor Street 40, 7400 Kaposvár, Hungary; Donko.Tamas@sic.medicopus.hu; 3Bábolna TETRA Ltd., Radnóti Miklós Street 16, 2943 Bábolna, Hungary; budaiz@babolnatetra.com (Z.B.); almasianita@babolnatetra.com (A.A.)

**Keywords:** pure lines, laying hen, beak trimming, feather pecking, plumage, body fat, egg production, mortality

## Abstract

**Simple Summary:**

The spread of both alternative and non-cage laying hen housing systems and the more forceful European refusal of beak trimming generate new problems in commercial egg production. The hybrid layers, which have been genetically selected under cage housing conditions for more decades, have lively temperament, are more susceptible for feather pecking and, in more cases, they are expressly aggressive, which led to permanent conflict situations in the large group keeping systems. Therefore, the omission of beak trimming could lead to an increased risk for feather pecking and consequently to a risk for increased mortality in the hen house by using the current commercial layers. Therefore, changes in the live weight, plumage and body condition, egg production, and mortality of different TETRA pure line non-beak-trimmed laying hens were compared during the egg-production period in our study, where the plumage condition was considered as an indicator trait for feather pecking. Our results confirm the findings of former studies that the genetic background of the hens is an important factor impacting feather pecking and suggest that breeding for an improved plumage condition might be a potential way to improve animal welfare in non-beak-trimmed layer flocks.

**Abstract:**

The experiment was carried out with altogether 1740 non-beak-trimmed laying hens, which originated from Bábolna TETRA Ltd., representing two different types (Rhode Island Red (RIR) and Rhode Island White (RIW)) and four different lines (Lines 1–2: RIR, Lines 3–4: RIW). The plumage and body condition of randomly selected 120 hens (30 hens/line) was examined at 20, 46, and 62 weeks of age. The egg production and the mortality of the sampled hens were recorded daily. Based on the results, it was established that the lines differ clearly in most of the examined traits. It was also pointed out that injurious pecking of the hens resulted not only in damages in the plumage but also in the body condition. The results obviously demonstrated that the highest egg production and the lowest mortality rate were reached by those hens, which had the best plumage and body condition. Because the occurrence of injurious pecking seems to depend on the genetic background, selection of the hens (lines, families, individuals) for calm temperament will be very important in the future in order to maintain the high production level in non-beak-trimmed layer flocks.

## 1. Introduction

As a result of the EU Council Directive 1999/74/EC of 19 July 1999, laying down minimum standards for the protection of laying hens, the use of the conventional cage system is prohibited since 1 January 2012 in the European Union member countries. Simultaneously with this, in some of the member countries (for example: Finland, Sweden, Denmark, Germany, and The Netherlands) the beak trimming of the laying hens—which is allowed till the 10th day of age in the European Union momentarily—is also prohibited, while in some other member countries its prohibition is under consideration.

The spread of both alternative and non-cage laying hen housing systems and the more forceful European refusal of beak trimming generate new problems in commercial egg production. The hybrid layers, which have been genetically selected under cage housing conditions for more than 70 years, have lively temperament, are more susceptible for feather pecking and, in more cases, they are characterized with aggressive behavior. This type of social stress was diminished in the small group (4–5 hens/cage) keeping systems, and its occurrence was efficiently reduced by trimming the beak with thermocauter, laser, or infrared light [1,2,3]. However, the aggression could cause permanent conflict situations in the alternative, large group keeping systems, where the ban of beak trimming might lead to an increased risk for feather pecking and consequently to a risk for increased mortality by using the current genetic stocks [1].

Because rearing is an important period for the development of behavioral patterns later in life, the effect of behavioral tests performed in this period was tested on the predictability of the plumage damage in adulthood [4]. However, because of the lack of significant correlations between the examined traits, it was established that the behavioral test reactions during the rearing period are not effective for the prediction of the plumage condition later in life.

In spite of this, comparing a classical animal model and a direct–indirect effects model for estimating genetic parameters for feather condition scores (FCS), it was pointed out formerly that methods of genetic selection that include indirect genetic effects offer perspectives to improve plumage condition in laying hens [5].

Although selection against feather pecking seems to be possible, it is necessary to characterize the genetic mechanism associated with this behavior [6]. In a recent paper [7], it was revealed that selection on high feather pecking leads to an increase of agonistic behavior. The correlation probably depends on the phase of establishing the social hierarchy in which the hens in a newly formed group are at the time of observation, and might disappear, after a stable ranking is evolved.

The aim of our study was on the one hand to compare the changes in the live weight, plumage and body condition, egg production, and mortality of different TETRA pure line laying hens during the egg-laying period and, on the other hand, to examine the effect of the plumage condition on the body condition of the hens, which has not been investigated so far. The aim of this study was not to make an ethological laboratory study but modeling the farm conditions, where the plumage condition was considered as an indicator trait for feather pecking.

## 2. Materials and Methods

### 2.1. Experimental Animals

The experiment was started with altogether 2600 non-beak-trimmed day-old pullets, which originated from the Bábolna TETRA Ltd. (Bábolna, Hungary), representing two different types (Rhode Island Red (RIR) and Rhode Island White (RIW)) and four different lines (Lines 1–2: RIR, Lines 3–4: RIW). Pullets originated from altogether 100 males (25 males per line) and 26 half sibling offspring per male were settled. The examined lines represent a part of the Rhode-type elite lines used by the Bábolna TETRA Ltd. for breeding brown egg layer hybrids. There are sire (Line 1 and Line 2) and dam (Line 3 and Line 4) lines between them, and they have different positions (A, B, C, D) in crosses. More detailed information about the selection philosophy of the pedigree lines can be read in the paper written by the geneticist of the Bábolna TETRA Ltd. [8].

### 2.2. Rearing Conditions

The pullets were raised up to 18 weeks of age in one closed building, in pens (using deep litter floor and 14 pullets/m^2^ stocking density) at the Experimental Poultry Farm of the Kaposvár University (legal predecessor of the Szent István University Kaposvár Campus), in Hungary. Pullets belonging to the same line were reared in the same pen. For lighting and feeding the recommendations of the TETRA-SL LL, parent stock management guide was applied.

### 2.3. Circumstances in the Laying House

At 18 weeks of age, 1740 pullets were moved to the closed laying house of the Experimental Poultry Farm of the Kaposvár University and were placed into EU-compatible, KOVOBEL SKNO-/30-60/ECS type furnished cages (10 birds/cage). Pullets in the same cage were reared in the same pen earlier. For lighting and feeding the recommendations of the TETRA-SL LL, parent stock management guide was applied.

For the examination of the plumage and body condition of the hens, 120 animals (30 per line) were randomly selected and assigned with wing tags individually at the time of moving to the laying house. The sampled hens were placed into 12 cages (3 cages per line), representing each line by one cage at each level of a three-level battery. The cages at the same level of the battery were placed beside each other, so the sampled hens could hear and see each other during the whole experimental period. Due to the random sampling, it may have occurred that some groups consisted of closely related individuals, but the chance of this was similar in all of the examined lines.

### 2.4. Examination of the Plumage Condition

The plumage condition of the sampled hens was examined at 20, 46, and 62 weeks of age. The plumage condition of all the sampled hens was controlled at five different body parts: neck, breast, wings, back, and tail. The plumage condition was evaluated on a 4-grade scale, where four points were given for the intact plumage and one point was given for the strongly damaged and incomplete plumage based on a reference photo series [9]. By summarizing the plumage points at the different body parts, a total plumage point was calculated for each hen, which could be ranged from 5 to 20. The scoring of the plumage condition was made by the same experienced scorer during the whole experimental period.

### 2.5. Examination of the Body Condition

The body condition (body fat content) of the sampled hens was determined at the same ages (20, 46, and 62 weeks) by means of computer tomography (CT) in vivo at the Institute of Diagnostic Imaging and Radiation Oncology of the Kaposvár University. Before the CT measurements, the live weight of the hens was always recorded.

During the CT scanning procedures, birds were fixed with belts in a special plastic container (Figure 1), without using any kind of anesthetic. Three animals were scanned simultaneously. Due to the special arrangement of the hens, they were separable on the CT images obtained (Figure 2).

The CT measurements consisted of overlapping 5 mm thick slices covering the whole body using a Siemens Somatom Sensation Cardiac 16 multislice CT scanner. From the images obtained the volume of fat was determined by using the total number of pixels with X-ray density values of fat, i.e., the range between −20 to −200 on the Hounsfield-scale. For determining the body condition of the hens, the ratio of the volume of fat to the live weight (relative body fat content) was calculated.

### 2.6. Measurement of Egg Production and Mortality

The egg production and the mortality of the sampled hens were recorded daily. The egg production was recorded per cages, while the mortality per treatment.

### 2.7. Statistical Analysis

For testing the normal distribution of the examined traits, the Shapiro–Wilk test was used. Where the data showed normal distribution (live weight and body condition), the One-Way Analysis of Variance (ANOVA) was performed for the statistical evaluation of the effect of line. Where the main effect of line was statistically proven (*p* < 0.05), the significance of between-group differences was controlled by the Tukey post hoc test. In those traits, where the data did not show normal distribution (plumage points and egg production), the non-parametric Kruskal–Wallis test was used for evaluating the effect of line. For testing the significance in the differences of the mortality rate between the lines, the Chi^2^-test was applied. Pairwise comparisons were made only at those ages when the main effect of line was significant (*p* < 0.05). All of these statistical analyses were performed by the 10.0 version of the SPSS statistical software package [10].

## 3. Results

### 3.1. Live Weight

The live weight of the experimental hens showed an increasing tendency in all of the examined lines as the age progressed (Table 1).

The relative increase in the live weight varied between 11.1% and 21.2%, reaching the lowest value in Line 1 and the highest in Line 3. At the end of the examined period, the average live weight of the hens in Line 3 and Line 4 was significantly higher than that of the hens in Line 1 and Line 2. The highest average live weight (observed in Line 3) was almost 200 g higher than the lowest in Line 2.

### 3.2. Plumage Condition

In spite of the live weight, the plumage condition of the hens showed a decreasing tendency with increasing age in all of the examined lines (Table 2).

The decrease of the plumage condition points varied between 1.1 and 3.7, reaching the lowest value in Line 3 and the highest in Line 1. Comparing these results with the changes in the live weight, it could be observed that a converse tendency exists between the increase of the live weight and the decrease of the plumage condition of the hens. The increase of live weight was the lowest, where the decrease of the plumage points was the highest (Line 1) and the increase of the live weight was found the highest, where the decrease of the plumage points was the lowest (Line 3).

Besides the highest decrease in the plumage points in Line 1, the absolute plumage points were also the worst in this line during the whole experimental period. The total plumage point of the hens in Line 1 differed significantly from that of the others already at the start of the egg production period (*p* < 0.05). While the hens in Lines 2–4 had almost fully intact plumage at 20 weeks of age (average total plumage points ranged between 19.4 and 19.7), the plumage was strongly damaged in the hens in Line 1 already at this early age (average total plumage point was 16.7). While total plumage points under 18 were not observed in Lines 2–4, seventeen hens in Line 1 had total plumage points between 11 and 17 and none of them had 20 points at this age.

Although the total plumage points declined in all examined lines during the experimental period, Lines 2–4 had relatively good plumage condition at the end of the experiment (average total plumage points varied between 17.4 and 18.6; Figure 3). In spite of this, the average total plumage point was only 13.0 in Line 1 at 62 weeks of age (Figure 4).

From Table 2, it is also visible that the hens in Lines 2–4 had better plumage conditions even at the end of the experiment than the hens in Line 1 at the beginning of the study.

When the plumage condition of the hens was examined on the different body parts separately, the best results were obtained in the region of the breast in all of the lines examined (Table 3).

In this body region, the plumage condition of the hens was totally complete in Line 2 and 3 during the whole experimental period, while only slight damages were observed on the hens in Line 1 and 4 at different ages. In this body region, no significant differences were found between the lines examined (*p* > 0.05).

At the other body parts (neck, back, tail, and wings), the plumage condition of the hens was significantly worse in Line 1 than in the other lines at all of the examined ages (*p* < 0.05). The plumage points of the hens showed a continuous decrease in these body regions with increasing age in Line 1, reaching the worst result on the neck at the end of the experiment.

The best results were obtained in Line 3, where the plumage was totally complete on the back, breast, and wings during the whole experimental period, while only slight damages were observed on the neck at 62 weeks of age and on the tail at all examined ages.

### 3.3. Body Condition

Similarly to the plumage condition, the body condition of the hens was also the worst in Line 1 during the whole experimental period (Table 4).

The decrease in the body fat content of the hens was 14.5% between the start and the end of the experiment in this line.

In spite of the other three lines, a continuous increase of the body fat content was observed in Line 3 during the whole examined period. The body fat content of the hens differed significantly from that of the others in this line as well as at 46 and 62 weeks of age. The relative body fat content of the hens in Line 3 was more than double than that of the hens in Line 1 at the end of the experiment.

From these results, it is well visible that the worst plumage and body condition were observed in the same line, namely, in Line 1, while the best plumage and body condition were observed also in the same line, namely, in Line 3.

### 3.4. Mortality

The mortality rate was the lowest in Line 3, where the plumage and body condition of the hens were found the best (Table 5).

In Line 1, where the plumage and body condition were the worst, the mortality rate was the second-highest and it differed significantly from that of Line 3.

The mortality rate was very similar in the first and second phases of the experiment in Line 2 and 3, while it was much higher in the second phase in Line 1 and 4. In line 4, the high mortality occurred despite the fact that the plumage of the hens was in relatively good condition even at the end of the experiment.

### 3.5. Egg Production

The egg production of the hens was the highest in Line 3 (Table 6), where the plumage and body condition were the best and the mortality rate was the lowest.

The egg production of the hens in Line 3 was significantly higher (by 38 eggs/caged hen) than that of the hens in Line 1, where the plumage and body condition were the worst and the mortality rate was the second-highest.

## 4. Discussion

The plumage condition of laying hens is an important indicator of their health and behavior. Therefore, various scoring systems [11,12,13] and various risk factors for feather damage have been already tested and described [14,15].

When the applicability of a 4-grade scale was tested by two independent experienced observers, highly significant correlations were found between the scores given by the two scorers [11]. Both observers ranked the average scores of the individual body parts’ plumage condition in the same order, but the significant tendency for one scorer to put higher points for tail- and wing plumage indicated that the use of the same scorer(s) if birds are scored at different ages within the same experiment is very important. Based on this finding, the scoring of the plumage condition of the hens was performed by the same experienced scorer at all examined ages in our study.

Because a summarized score of different body parts—similarly to our experiment—was often used to describe the overall condition of the plumage of a bird, it was also tested formerly, which information could be lost when summarizing the scores of the separate body parts [16]. By testing two models, where the first model included the whole body score as one outcome variable and the second model the scores of the individual body parts as multiple outcome variables, it was established that the investigation of the individual body parts allowed for consideration of the influences on each body part separately and for the identification of additional influences. Furthermore, ambivalent influences could be detected with this approach, and possible dilutive effects could be avoided.

Supporting these findings, it was pointed out in our experiment that the use of the summarized plumage score can cover up some differences or similarities in the different body regions. While the total plumage point showed significantly worse plumage condition in Line 1 compared to the others in our study, the detailed examination of the different body parts clearly demonstrated that it is only true for the region of the neck, back, tail, and wings, and not for the region of the breast.

Our results also proved that the more serious damages in the plumage of the hens mainly occur on those body parts, which are easily accessible for feather pecking. The region of the breast, which is mostly hidden from this unfavorable behavior, showed only slight or no damages in the examined lines.

However, when laying hens were selected for and against feather pecking in a former trial, the plumage condition was found better in the low feather pecking group on all of the examined body parts (neck, breast, back, wings, and tail) already in the 3rd generation [17]. In this study, it was also revealed that the feather pecking behavior in adult hens was significantly higher in the high feather pecking than in the low feather pecking group. The proportion of hens recorded feather pecking in a 180 min observation period was 75% and 49% in the high and low feather pecking groups, respectively.

When the heritability of feather pecking was examined in a social test in a later study, the h^2^ values were 0.12 for gentle feather pecking and 0.13 for ground pecking at 6 weeks of age and 0.15 for gentle feather pecking and 0.30 for ground pecking at 30 weeks of age [18]. These results clearly demonstrated that selection seems to be a possible way against feather pecking.

Because foraging and fear are considered as an important causal motivator for feather pecking, the relationships between severe feather pecking (bouts of severe feather pecks delivered), foraging (sum of walking and litter pecking), and open-field activity (number of steps) were tested in birds selected for high or low levels of feather pecking [19]. Because the genetic correlations between the examined traits were very low or not measurable, it was established that foraging and open-field activity do not have a causal influence on severe feather pecking despite the hypothesis.

When the relationship between feather pecking, feather eating, and general locomotor activity was examined on a similar population (lines selected for high or low levels of feather pecking), it was pointed out that an increase in feather eating leads to an increased feather pecking and that a raise in general locomotor activity results in a higher feather pecking value [20].

In a recent study, it was demonstrated that the birds selected on high feather pecking had more active responses (i.e., they approached a novel object sooner, vocalized sooner and more, showed more flight attempts, and had shorter tonic immobility durations) than the non-selected control birds or birds selected on low feather pecking, at both young and adult ages [21].

Because fear is frequently reported to be related to feather pecking, the quantitative genetic analysis of feather pecking and three fear test traits were carried out in a former study in laying hens [22]. Results of this experiment suggest that tonic immobility measured early in life is moderately correlated with feather pecking, therefore, it seems to be beneficial to use it as a breeding criterion to reduce feather pecking of the hens.

In former studies, findings also indicated that selection for feather pecking can lead not only to changes in behavior but also in egg production of laying hens. In a recent study, authors showed that selection might help to reduce feather pecking, but this might result in an unfavorable correlated selection response reducing egg production [23]. In contrast, egg production and also feed efficiency was found better in the line selected for low feather pecking than in the line selected for high feather pecking in another study [24].

Our results seem to confirm this latter finding, but no clear reduction in the egg production was detected with decreasing plumage condition of the hens. The differences in the egg production found in our study could be rather due to the differences in the genetic background of the examined lines.

Because the number of examined hens was limited in our study, larger datasets are needed to verify the correlation between the plumage condition and egg production in non-beak-trimmed layer flocks.

While nutritionists assume that nutritional deficiencies are the main causes of feather pecking and cannibalism, ethologists consider them as a result of a fundamental drive related to feeding behavior. From a review paper [25], it is known that the nutrient deficiencies increase the exploratory behavior, which can be directed towards the feather cover of group mates. However, it is also known from this paper that not only the composition but also the structure of the feed can influence the behavior of the birds. It seems that there is a consistent trend of the feed structure with coarse particles, especially pellets and crumbles, producing more feather pecking and cannibalism than finely ground feed. This effect is explained by the influence of feed structure on the time spent feeding. Pelleted or coarsely ground feed reduces the time required to ingest the feed, and thus, may not allow to fulfill the drive of food pecking and exploration.

Because both the structure and composition of the feed were the same for all of the examined lines in our experiment, the differences observed in the plumage condition cannot be attributed to nutritional factors.

Feather pecking became the most prevalent behavioral problem in laying hens since 1 January 2012 due to the ban on conventional cages in the European Union [26]. The search for alternative housing systems, which are more animal-friendly, led to the development of small group housing systems. However, in former studies, it was pointed out that the group size had a significant effect on the plumage condition of the hens. The plumage damage of single birds was always less than that of birds kept in groups [13], while in group housing systems, the plumage condition was always better in smaller groups [27].

In former experiments, it was also stated that the environmental enrichment did not have any unambiguous positive effect on the plumage condition of the hens neither in cages [28,29] nor in non-cage systems [4]. Results demonstrated that the furnished elements of the small group housing systems may improve the welfare of the hens, but these elements further may force feather pecking behavior by giving the hens an incentive to peck [30].

Since the hens were kept in equally equipped furnished cages in our experiment, the keeping system cannot be responsible for the differences observed in the plumage condition of the hens likewise.

For improving the adaptability and well-being of layers in large multiple-bird cages, the applicability of the “group selection” was tested in a former study [31]. With this procedure, each sire family was housed as a group in a multiple-bird cage and selected or rejected as a group. In this study, the annual percentage mortality of the selected line in multiple-bird cages decreased from 68% in the second generation to 8.8% in the sixth generation and so the percentage mortality of the selected line in multiple-bird cages decreased to the level of the unselected control in one-bird cages (9.1%). The similar survival of the selected line in multiple-bird cages and the control in one-bird cages suggests that beak-trimming of the selected line would not further reduce mortalities, which implies that group selection may have eliminated the need to beak-trim. This finding seems to be very important because besides the ban on conventional cages in the European Union, the ban on beak trimming in some member countries of the EU was the other main reason for feather pecking and cannibalism, becoming the most prevalent behavioral problem in laying hens. It was proven in several former experiments that birds with untrimmed beaks had significantly more plumage damage at the end of the rearing period as well as during the whole laying period than birds with trimmed beaks [4,15,32]. In one of these studies [4], it was established that only 5.2% of the beak-trimmed hens had feather loss or wounds at 43 weeks of age, while this ratio was 72.9% in the non-beak-trimmed hens at the same age.

The ban on beak trimming could increase also the mortality in the hen house by using the current genetic stocks [1]. Confirming this statement, very high (>16%) mortality was observed in three of the four examined lines in our experiment. Because the general health status of the hens was normal and infections and other diseases did not occur in our study, the high mortality rate can be attributed to the lack of beak trimming.

Besides the environmental (welfare) conditions, the production of the laying hens depends also on their body condition during the laying period. It was already pointed out formerly that the hens selected for increased egg production, became smaller, and have less body fat compared to the earlier strains [33].

It has also been shown in a former study that the body condition of laying hens could be very different at the end of the laying period [34]. It was shown that 7%, 56%, 35%, and 2% of the hens had a body condition score of 0, 1, 2, and 3 at the end of the experiment, respectively. In some of the birds with a body condition score of 0, no dissectible body fat was observed. However, the empty body weight apparently increased with increasing body condition score, and on average the birds with a body condition score of 3 were over 50% heavier than the birds scoring 0.

Although the body condition scoring method was found to be useful for predicting the bird’s body reserves, more accurate in vivo techniques were developed and used for this purpose. One of these techniques is the X-ray computer tomography (CT), which was effectively used in former experiments for the in vivo determination of the body composition in different poultry species [35,36,37]. In the case of laying hens, it was already used for following changes in the body composition of different genotypes during the egg-laying period [38], for the examination of the effect of initial body fat content on egg production [39], for following changes in the body composition during forced molting [40], and for the determination of bone density and breaking strength [41,42]. However, it was not used for the examination of the relationship between the plumage and body condition formerly, which was demonstrated in the present study.

Although the number of examined hens was limited due to the expensive in vivo body composition analysis in this study, the results suggest that the incidence of feather pecking has more than just animal welfare implications. However, to confirm these results, further investigations are needed involving a larger number of experimental animals, and more detailed data of the studied hens (pedigree, genomic) is required.

## 5. Conclusions

In the present study, it was demonstrated that the ban of beak trimming can lead to different plumage damages in the laying hens originating from different genetic lines. The results also clearly indicated that the more serious damages in the plumage of the hens mainly occur on those body parts, which are easily accessible for feather pecking.

Based on the results, it was established that the occurrence of feather pecking in non-beak-trimmed laying hens results not only in damages in the plumage condition but also in decreasing body fat content, i.e., in inferior body condition. The results clearly demonstrated that the best plumage and body condition resulted in the highest egg production and in the lowest mortality rate.

As the main drivers for differences in the plumage condition and health and production traits were ruled out through the study design, the differences were very likely due to the line. So, the results underline the findings of other studies where the genetic background of the hens was found to be a factor impacting feather pecking. Our findings suggest that breeding for an improved plumage condition (which is an indicator for low feather pecking) might be a potential way to improve animal welfare in non-beak-trimmed layer flocks. However, larger datasets and many more information regarding the genetics of the lines will be necessary to really prove that.

## Figures and Tables

**Figure 1 animals-11-00500-f001:**
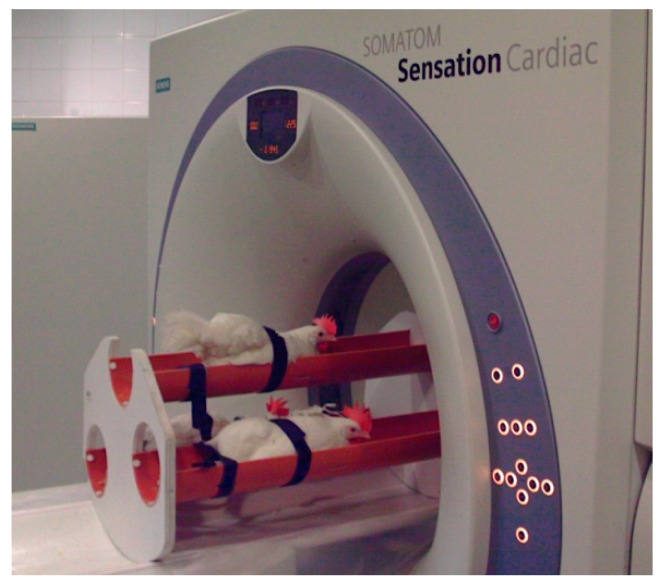
Laying hens in a special plastic container.

**Figure 2 animals-11-00500-f002:**
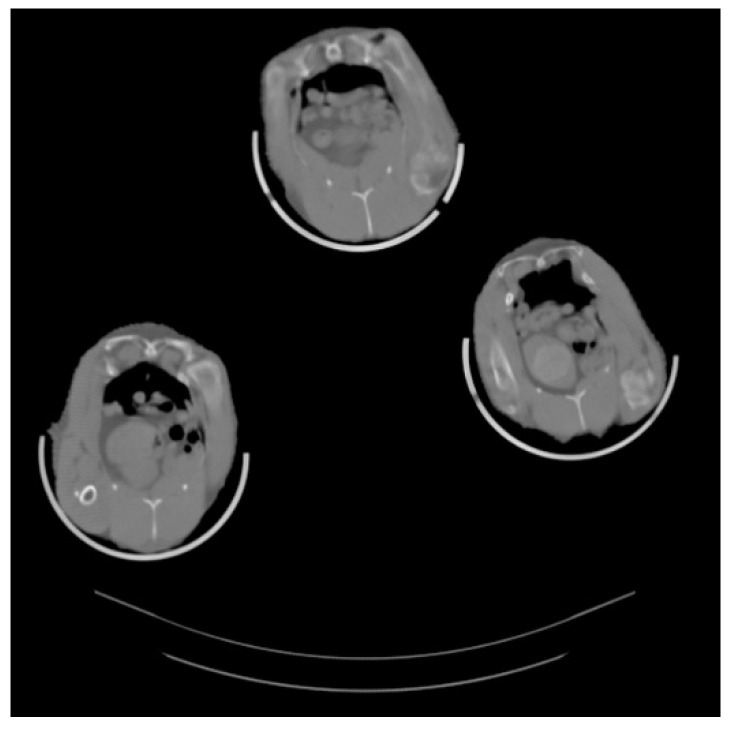
Cross-sectional CT image about laying hens.

**Figure 3 animals-11-00500-f003:**
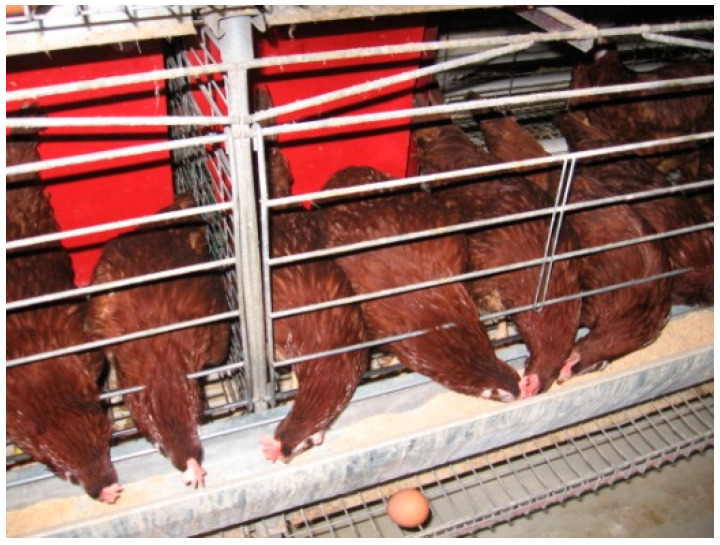
Good plumage condition in Line 2 at 62 weeks of age.

**Figure 4 animals-11-00500-f004:**
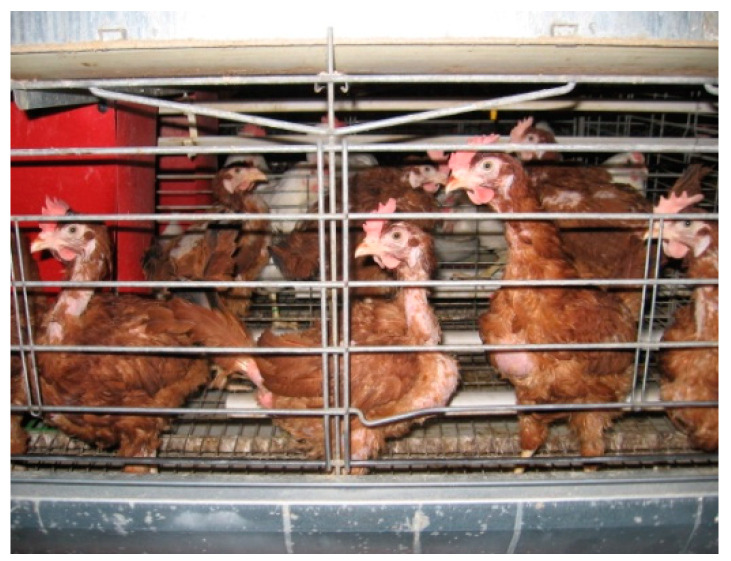
Bad plumage condition in Line 1 at 62 weeks of age.

**Table 1 animals-11-00500-t001:** Effect of genetic background on the live weight of non-beak-trimmed pure line laying hens.

Age (Weeks)	Live Weight (g)							
	Line 1		Line 2		Line 3		Line 4	
	*n*	Mean ± SD	*n*	Mean ± SD	*n*	Mean ± SD	*n*	Mean ± SD
20	30	1559 ^b^ ± 81	30	1473 ^a^ ± 74	30	1568 ^b^ ± 66	30	1577 ^b^ ± 65
46	27	1703 ^a^ ± 121	28	1650 ^a^ ± 125	30	1826 ^b^ ± 185	26	1705 ^a^ ± 128
62	21	1732 ^a^ ± 152	26	1705 ^a^ ± 132	29	1901 ^b^ ± 145	15	1831 ^ab^ ± 150

^a,b^ Different letters in the same row indicate significant differences (*p* ≤ 0.05).

**Table 2 animals-11-00500-t002:** Effect of genetic background on the plumage condition of non-beak-trimmed pure line laying hens.

Age (Weeks)	Total Plumage Point *							
	Line 1		Line 2		Line 3		Line 4	
	*n*	Mean ± SD	*n*	Mean ± SD	*n*	Mean ± SD	*n*	Mean ± SD
20	30	16.7 ^a^ ± 1.6	30	19.4 ^b^ ± 0.9	30	19.7 ^b^ ± 0.5	30	19.5 ^b^ ± 0.7
46	27	15.3 ^a^ ± 2.5	28	18.4 ^b^ ± 1.0	30	19.3 ^b^ ± 0.5	26	19.1 ^b^ ± 1.1
62	21	13.0 ^a^ ± 2.2	26	17.4 ^b^ ± 1.8	29	18.6 ^b^ ± 0.9	15	17.5 ^b^ ± 2.2

* Total plumage point could be ranged from 5 (worst) to 20 (best). ^a,b^ Different letters in the same row indicate significant differences (*p* ≤ 0.05).

**Table 3 animals-11-00500-t003:** Effect of genetic background on the plumage condition of non-beak-trimmed pure line laying hens at different body parts.

Age (Weeks)	Plumage Point *							
	Line 1		Line 2		Line 3		Line 4	
	*n*	Mean ± SD	*n*	Mean ± SD	*n*	Mean ± SD	*n*	Mean ± SD
Neck								
20	30	3.00 ^a^ ± 0.32	30	3.91 ^b^ ± 0.32	30	4.00 ^b^ ± 0.00	30	4.00 ^b^ ± 0.00
46	27	2.64 ^a^ ± 0.75	28	3.18 ^b^ ± 0.88	30	4.00 ^c^ ± 0.00	26	3.86 ^c^ ± 0.47
62	21	1.65 ^a^ ± 0.93	26	2.90 ^b^ ± 0.88	29	3.20 ^b^ ± 0.45	15	3.45 ^b^ ± 0.69
Back								
20	30	3.57 ^a^ ± 0.73	30	4.00 ^b^ ± 0.00	30	4.00 ^b^ ± 0.00	30	4.00 ^b^ ± 0.00
46	27	3.41 ^a^ ± 1.00	28	4.00 ^b^ ± 0.00	30	4.00 ^b^ ± 0.00	26	4.00 ^b^ ± 0.00
62	21	2.75 ^a^ ± 0.85	26	3.80 ^b^ ± 0.63	29	4.00 ^b^ ± 0.00	15	3.82 ^b^ ± 0.60
Tail								
20	30	2.87 ^a^ ± 0.69	30	3.55 ^b^ ± 0.53	30	3.67 ^b^ ± 0.55	30	3.58 ^b^ ± 0.51
46	27	2.45 ^a^ ± 0.69	28	3.36 ^b^ ± 0.48	30	3.33 ^b^ ± 0.55	26	3.50 ^b^ ± 0.51
62	21	2.10 ^a^ ± 0.79	26	3.40 ^b^ ± 0.52	29	3.40 ^b^ ± 0.55	15	3.00 ^b^ ± 0.45
Breast								
20	30	3.91 ± 0.22	30	4.00 ± 0.00	30	4.00 ± 0.00	30	4.00 ± 0.00
46	27	3.91 ± 0.22	28	4.00 ± 0.00	30	4.00 ± 0.00	26	4.00 ± 0.00
62	21	4.00 ± 0.00	26	4.00 ± 0.00	29	4.00 ± 0.00	15	3.82 ± 0.60
Wings								
20	30	3.35 ^a^ ± 0.61	30	3.91 ^b^ ± 0.32	30	4.00 ^b^ ± 0.00	30	3.88 ^b^ ± 0.41
46	27	2.86 ^a^ ± 0.69	28	3.82 ^b^ ± 0.42	30	4.00 ^b^ ± 0.00	26	3.82 ^b^ ± 0.51
62	21	2.45 ^a^ ± 0.69	26	3.30 ^b^ ± 0.48	29	4.00 ^c^ ± 0.00	15	3.36 ^b^ ± 0.67

* Plumage point could be ranged from 1 (worst) to 4 (best). ^a,b,c^ Different letters in the same row indicate significant differences (*p* ≤ 0.05).

**Table 4 animals-11-00500-t004:** Effect of genetic background on the body condition of non-beak-trimmed pure line laying hens.

Age (Weeks)		Relative Body Fat Content (cm^3^/kg)						
		Line 1		Line 2		Line 3		Line 4
	*n*	Mean ± SD	*n*	Mean ± SD	*n*	Mean ± SD	*n*	Mean ± SD
20	30	50.5 ^a^ ± 11.2	30	62.8 ^b^ ± 9.8	30	61.8 ^b^ ± 19.8	30	57.2 ^ab^ ± 15.4
46	27	52.7 ^a^ ± 13.2	28	62.0 ^a^ ± 20.3	30	81.7 ^b^ ± 21.5	26	65.9 ^a^ ± 24.5
62	21	43.2 ^a^ ± 16.0	26	56.7 ^a^ ± 15.7	29	94.8 ^b^ ± 28.3	15	62.7 ^a^ ± 24.0

^a,b^ Different letters in the same row indicate significant differences (*p* ≤ 0.05).

**Table 5 animals-11-00500-t005:** Effect of genetic background on the mortality of non-beak-trimmed pure line laying hens.

Age (Weeks)	Mortality							
	Line 1		Line 2		Line 3		Line 4	
	*n*	%	*n*	%	*n*	%	*n*	%
20–46	3	10.0	2	6.6	0	0.0	4	13.3
46–62	6	20.0 ^bc^	2	6.6 ^ab^	1	3.3 ^a^	11	36.7 ^c^
20–62	9	30.0 ^bc^	4	13.2 ^ab^	1	3.3 ^a^	15	50.0 ^c^

^a,b,c^ Different letters in the same row indicate significant differences (*p* < 0.05).

**Table 6 animals-11-00500-t006:** Effect of genetic background on the egg production of non-beak-trimmed pure line laying hens.

Age (Weeks)	Egg Production (Eggs/Housed Hen)							
	Line 1		Line 2		Line 3		Line 4	
	*n*	Mean ± SD	*n*	Mean ± SD	*n*	Mean ± SD	*n*	Mean ± SD
20–46	3	115 ^ab^ ± 11	3	104 ^a^ ± 10	3	133 ^c^ ± 8	3	127 ^bc^ ± 7
46–62	3	52 ^a^ ± 15	3	53 ^a^ ± 13	3	72 ^b^ ± 12	3	63 ^ab^ ± 9
20–62	3	167 ^ab^ ± 25	3	157 ^a^ ± 12	3	205 ^c^ ± 18	3	190 ^bc^ ± 15

^a,b,c^ Different letters in the same row indicate significant differences (*p* ≤ 0.05).

## Data Availability

The data presented in this study are available on request from the corresponding author. The data are not publicly available due to sensitive data about some pedigree lines.

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
