# Peer review of "Comparison of Changes in the Plumage and Body Condition, Egg Production, and Mortality of Different Non-Beak-Trimmed Pure Line Laying Hens during the Egg-Laying Period"

_animals, 2021, doi:10.3390/ani11020500_

Round 1
Reviewer 1 Report
- Feather pecking, even within the same genotype, has a large variation. For determining feather pecking only 30 birds per genetic line are used, placed in 3 cages per line, so these 30 hens are not independent from eachother. So basically n=3. For feather pecking really this is very far from an adequate number. Also, I miss a good description of the model used.
- Knowledge of (causation of) injurious pecking is very poor and the paper should be revised thoroughly on this! Some examples:
- ‘aggressive behaviour’ this is wrong wording. I take it the authors mean injurious pecking. This however mostly is NOT aggression, aggressive pecking is directed at the head, injurious pecking is directed to other body parts and has a different underlying motivation, no aggression! This wording is used throughout the paper and makes the paper to me written by people not having knowledge of laying hen behaviour.
- the fact that a general scoring list is used, lacking the cloaca area and not distinquishing back of head, back of neck and front of neck, makes that this paper lacks some essential information.
- rearing pullets on litter and then placing them in cages for the production period is asking for problems regarding injurious pecking
- The fact that the lowest body weight was connected with the worst feather condition can not be concluded in this setup and with this limited number of hens!
- I indicated 'not applicable' for improving the research design and conclusion supporting the results, because with this limited number of birds, you just need to redo the whole experiment, there is no way to improve the design of the work done or draw any concluysions.
Therefore I can't do much more than reject this paper.
Reviewer 2 Report
The title and purpose of the study should be appropriate to the content. Examination of the effect of genetic background announces genetic analyses that are not included in the work.
The experiment was carried out with non-beak-trimmed laying hens, representing four different lines. Questions and comments:
- What are the differences in breeding objectives between lines?
- Is the average relationship of the individuals making up the test groups within the pure lines known?
For the statistical evaluation of the effect of genotype the one-way ANOVA was used.
- Was the effect of the cage battery level on the analysed features verified?
- Do the examined traits have a normal distribution?
- Level of significance (p-value) in the tables is redundant.Lettering the mean differences will suffice.
- In the case of plumage condition, a non-parametric test should be used.
- The significance of the between-group differences should be verified e.g. by Tukey's test.
Reviewer 3 Report
Dear authors,
my comments aim to improve your manuscript and I encourage you to make the necessary changes to make the manuscript publishable.
Comments:
Lines 67-68: There is a brand-new paper on that scope (see https://doi.org/10.1016/j.applanim.2020.105177). If possible, rather use more recent literature to cite.
Line 79: 2 lines from each of the two breeds were chosen. What are the differences between the lines within one breed/type? Were the divergently selected for a certain trait? Why were they taken? Please clarify.
Line 81: Were all pens in the same building? If so, please replace ‘in a closed building’ by ‘in one closed building’.
Lines 87: 1,740 animals were in the study. Why did you sample only such a small number of individuals (30 per line)? Were the other animals not phenotyped? Normally larger sample sizes lead to more reliable results…
Lines 124-125: The egg production was recorded per cage. The mortality per line was up to 30 %. I wonder how many individuals were left for the recording of the traits at the 3rd timepoint of measure as some of the individuals died before…
Additionally, was the mortality rate the ratio of hens that died within a certain period and the total number of hens per line? If so, was the total number of hens the actual number of hens at each timepoint (which means a decreasing number as some were dying during the experiment)?
Lines 127-132: This section needs to be rewritten. More (detailed) information is required for the reader to understand correctly what you’ve done to evaluate the dataset and to interpret the results.
Were the observed phenotypes normally distributed (did you test that?) or were some transformations required? Did you have to care about some other fixed effects or maybe pre-corrected for them? What about the homogeneity of variances? This is important for the fulfillment of the prerequisites to apply ANOVA and as no descriptive statistics are given in the paper, this information is missing.
Please indicate that LSD and Chi² tests were only performed for the results where the differences between the lines were significant (because abcd indications are missing in the result tables without explanation…). What was the maximal p-value you considered to prove significant differences between lines? (0.01??) Give information!
Please replace the word ‘genotype’ by ‘line’ which is more appropriate as no genetic information but the line per se was included in the study.
Why did you use Chi² test for mortality? Why are no S.E. given in the result tables for this trait? Please explain.
Lines 134-227: Please add the number of individuals at each time point to each of the tables. This helps the reader to better interpret the results. Further, adding the standard deviations to the means (which were shown in the tables) would allow to infer the within line variances. I think that is necessary du to the limited sample size were the observation of a single individuals may strongly influences the mean, i.e. outliers may rather compromise the interpretation of the mean value than compared with in bigger samples.
You provide S.E. in your tables. Please indicate how they were calculated. Why do you provide this information but not discuss it later?
Do you have more comparable pictures for figures 3 and 4, i.e. both figures showing the front or the back of the hens?
Lines 270-274: Other studies had different results. Maybe the following paper is beneficial for this discussion point (see Poult Sci. 2014 Apr;93(4):810-7. doi: 10.3382/ps.2013-03638.).
Discussion (lines 229-335)
In general:
I think the word ‘established’ is not always appropriate and correctly used. Some expressions are used very often which makes the discussion boring to read from a grammar point of view.
The discussion would benefit from the inclusion of studies investigating the heritability of feather pecking traits and the meaning of such studies for your results (e.g. Lutz et al 2015, Iffland et al 2019). This would support your findings concerning the differences between the lines.
Please explicitly point out that you have ruled out the influence of previously studied factors that may impact the plumage condition and hence you infer that the differences are due to different lines.
The results were not discussed from a statistical point of view. However, it would help the reader to better interpret the results and draw inferences on what should be taken into account or can be made better in follow-up studies in this field. Such a section should be added.
Round 2
Reviewer 1 Report
Line 13&48: I don’t understand “and the more forceful European refusal of beak trimming”. What does it refer to? There is no European law that bans beak trimming
Line 14-15: “hybrid layers, which were improved in cages for more than 70 years, have vital temperament, are susceptible for feather pecking and in more cases they are expressly aggressive”
- ‘were improved’ should be replaced by ‘have been genetically selected’
- What is ‘vital’ temperament?
- ‘susceptible for feather pecking’ this in fact is incorrect, there are large differences and for instance white genotypes are not as susceptible as brown ones
- ‘expressly’ is a strange wording and if ‘typically’ is meant, see my previous commend
- ‘aggressive’ this is wrong wording as feather pecking is NOT motivated by aggression, neither is cannibalism.
Line 16-17: “the leaving of 16 beak trimming could increase the mortality”
- It should be the omission or absence of beak trimming
- Not trimming as such does NOT increase mortality, it only leads to an increased risk for feather pecking and consequently a risk for increased mortality
Line 22-23: “therefore selection seems to be an effective tool to solve the current problems in the non-beak-trimmed layer flocks”
- There are many papers showing the effect of selection on feather pecking. Klaer had selected lines over many generations (Kjaer, J.B., Hocking, P.M., 2004. The genetics of feather pecking and cannibalism, in: Perry, G. (Ed.), Welfare of the Laying Hen, Cabi Publishing, Cambridge, pp. 109-121.). So if this is the conclusion of your paper it is not very new information
- What is meant with ‘current problems’ this is a vague term
Line 31-32: “The results clearly demonstrated that the highest egg production and the lowest mortality rate were reached by those hens, which had the best plumage and body condition.
- Again this really is not very new information
Line 50-51: “they are characterized with aggressive behaviour”
- Wrong statement, see earlier remark
Line 53: “agressivity”
- Wrong statement, see earlier remark
Line 54: “where the 54 ban of beak trimming could increase the mortality”
- The ban as such did not increase mortality, it may have led to an increased mortality. This type of sloppy formulations are not very scientific!
Line 68-71: this text is literally copied from the referred paper!
Line 94: I do hope each line was in all 3 levels! I miss a statement for this.
Line 104: I do know the Tauson method, which is a general method to assess feather quality. However, not all feather damage is caused by feather pecking, so for assessing feather pecking one should use a different method, e.g. the method presented in the Welfare Quality protocol.
Line 136: ‘mortality individually’ Is there any mutual mortality??? This expression is strange. Does it mean you count the number of dead hens per treatment? How did you analyse this? Birds in one cage are not independent, so your experimental unit is a cage.
Line 126: this statistical paragraph is insufficient. What is the experimental unit? What models are used? Thre are lines, but also cages and cage levels, so some kind of blocking needs to be done.
Line 170: 4 lines with different growth. They also differ in feather condition. The setup of the experiment is insufficient to find any connection between these two traits! For that you need a within-lines comparison.
Line 234: how is mortality rate calculated? Based on total mortality per group or based on average mortality% per cage? Mortality is a very variable trait, so the number of birds per group in this study is too low to draw any conclusion.
Line 240: is this egg production per hen housed or per hen present? I would say per hen present. This should be indicated!
Line 277-293: why is this included if the authors didn’t do any behavioural observations? This can be left out!
Line 301-311: why is this included if the authors didn’t do any feed-experiment? This can be left out!
Line 315-316: this ref is to the legislation, I miss proof of this statement and I think there is no proof!
Line 315-328: can be left out, it is enough to state the birds were housed in similar circumstances.
Line 376: this completely ignores typical bird behaviour. Not all easily accessible body parts are pecked at for the same reason. Feather damage to the back of the head is aggressive behaviour, to lower parts of the neck is feather pecking.
There is a lot of information in the paper that really is not new. The design of the study is very poor. I would strip this paper to only the body composition part.
Reviewer 2 Report
The differences between the lines in the examined traits may be explained by a different breeding goal and selection history. However, I am aware of the breeding secrets of Bábolna TETRA.
Was it possible to verify the random selection of individuals forming the experimental groups? Theoretically, the group may consist of closely related individuals (FS or HS), which may affect the objectivity of the test results. It may be an advantage for groups or a disadvantage for individual service checks.
The numbers, mean and standard deviation designations are missing from the header of the tables. Designation of the count (n =) in the tables is redundant, as well as ± (plus-minus sign). The designation of P <0.05 should include the sign ≤ (less than or equal to)
Reviewer 3 Report
In general:
The study design and the number of studied animals is not sufficient to infer whether breeding against feather pecking or hens with a calm temperament is possible or not.
In my opinion, the following point should be clearly pointed out in the manuscript:
- Neither feather pecking nor traits regarding the temperament were measured in the study. The plumage condition was considered as an indicator trait for feather pecking.
- In the response to the comments you pointed out that the effect of the plumage condition on the body condition was one major research goal and has not been investigated so far. In terms of this research goal, the number of animals in the study might be sufficient. You should emphasize this in the manuscript.
- (Significant) differences between the lines were observed for the traits investigated. A good plumage condition came along with a good body condition and a low mortality. As the main drivers for differences in plumage condition and the health and production traits were ruled out through the study design, the differences were very likely to be due to the line (which was also shown by your statistics). So the results underline the findings of other studies where the genetic background of hens was found to be an factor impacting feather pecking. Your findings suggest that breeding for an improved plumage condition (which is here an indicator for low feather pecking) might be a potential way to improve animal welfare in non-beak-trimmed flocks. Larger datasets and many more information regarding the genetics of the lines will be necessary to really prove that.
Lines 80-82
There is still too less information about the lines. Where are the differences between the lines (e.g. selected as sire lines or dam lines,…). You responded to the comment but did not mention in the manuscript… Were the pullets’ parents a randomly chosen sample from the respective lines’ populations? Is there any information about the population structure of the birds in the experiment within lines? E.g. number of parents they descend from?
This section is essential for the study as you came to the conclusion that lines differ and hence breeding could improve animal welfare etc. for non-beak-trimmed stocks…
Line 90
You should mention in the manuscript why only 30 birds per line were chosen. Maybe it would help if you emphasized the goal to investigate the effect of the plumage condition on the body condition in the introduction.
Lines 221-227
Please describe the results more in detail. E.g. point out that the by far highest mortality rate was found in line 4 in the second period although their plumage condition was very good according to the results ‘plumage conditions’. This should also be part of the discussion.
Lines 229-236
For the interpretation of the results it is important to consider the number of eggs given the number of hens from which the eggs were produced. When looking at the mortality rate, line 1 and 3 are not too bad. Did you consider that when comparing the egg numbers and infer significance? I doubt that the egg production per hen was significantly different. This should then also be part of the discussion.
Discussion:
The discussion is in parts too general. I miss a direct discussion of the results obtained from the present study (e.g. see the points that were mentioned in the comments concerning the results). The discussion should be more precise and also focusing the interpretation of the results, not only comparing the results with other studies.
Additionally:
Here the citation you asked for (I forgot that the name was Grams at this time)
Quantitative genetic analysis of traits related to fear and feather pecking in laying hens
Vanessa Grams*, Stefanie Bögelein*, Michael A. Grashorn*, Werner Bessei*, Jörn Bennewitz
doi: 10.1007/s10519-014-9695-1
Genetic parameters and signatures of selection in two divergent laying hen lines selected for feather pecking behaviour
Vanessa Grams, Robin Wellmann, Siegfried Preuß, Michael A. Grashorn, Jörgen B. Kjaer, Werner Bessei & Jörn Bennewitz
https://doi.org/10.1186/s12711-015-0154-0
Round 3
Reviewer 3 Report
Dear Authors,
the manuscript has been greatly improved. There are still some minor comments that should be adressed. Please find them in the following:
Generally:
Language correction is required. The style of writing should be improved. E.g. there are many phrases repeatedly used at high frequency, the word established is still often not used correctly, …
Lines 19-22:
The aim of the study in the Simple Summary should hit the aim of the study described in the Introduction. Plumage condition (as indicator trait for feather pecking) was not mentioned although the study is mainly about this.
Line 23:
replace ‚which‘ by ‚and‘
Lines 76-77:
Replace ‘partly’ by e.g. ‘on the one hand,… on the other hand’ that fits better
Lines 85-91:
There is still only few information on the genetic background of the lines from a breeders’ perspective. It was interesting for the interpretation of the results which lines are the sire lines and which are the dam lines. Just indicate them. Were they selected for different traits? Aren’t there any papers or other sources to cite that give further information? If yes, please cite.
Line 385
Replace ‚genotypes‘ by ‚lines‘ to be consistant
Lines 397-399:
This was not shown in the study! Plumage condition was the indicator trait for feather pecking, no traits regarding the temperament of individuals were measured.
Related to this comment: It must be made clear for the readers that the plumage condition links your findings to feather pecking (which is indeed also correlated with the temperament). Please consider this throughout the manuscript and adjust where required.
Comments on the discussion:
The discussion is somehow boring to read as many phrases were repeatedly used at high frequency.
The link between the plumage condition and feather pecking is missing. I know that this seems obvious but please explain and cite a study that investigated the relationship between these traits. Important measures to mention would be the (phenotypic and/or genetic) correlation between the traits (link between the traits). Further, mentioning heritabilities for these traits would help the reader to assess how much genetics influence the traits (link between these traits and the genetics of animals).
The relationship between behavior traits and feather pecking becomes not clear although many studies are cited. Please rewrite this section. Do not give too many details but rather take the extracts of the study to establish a red thread in your discussion. Are birds that peck more also more active? Are the results of those studies in line with your statements about calm behavior and breeding?
Round 4
Reviewer 3 Report
Comments to the Authors
Lines 20-23
The aim of the study described in the simple summary cannot be examined in your investigations. Just describe the aim as it was described in the introduction because this is what you’ve shown and what the results can tell us from this study. Again, the genetic background is more complicated and traits that point on the behavior were not observed. The study just showed that there were differences between the lines regarding the traits observed. This suggests that genetics is a factor which (among others) influences the plumage condition and, as it was used as an indicator for feather pecking, might also influence feather pecking. How and to what extent genetics plays a role regarding the traits observed can not be determined by using your study design.
Lines 90-91:
Please be more explicit. Which lines were the sire and dam lines? You gave a more detailed information in the response to the reviewers, please also give that information in the manuscript. This is also improtant as there were no English versions of the references available (or I just couldn't find using the present citation format...).
Table 3:
Please correct the decimal separator which should be a point and not a comma.
Lines 418-419
However, to confirm these results further investigations seem to be needed involving a larger number of experimental animals and more detailed data of the studied hens (pedigree, genomic) is required.
This must be added.
(Such information would be necessary to infer what makes the differences between the lines and whether the differences were just observed for the sample or the differences are also valid for the entire populations of the lines. As no further information on the hens studied were given, it is not possible for the reader to assess whether the samples were random samples of the population or (by chance) a group of related individuals, i.e. one or few families from the population.)
If you do have more information on the hens you should mention this in the Material and Methods section. But still the number of hens would not be sufficient to really prove genetic differences…
Line 421-428:
However, since there are only a few egg hybrid breeding companies in the world, we think that it is a huge thing that samples from the TETRA elite lines were investigable in this experiment thanks to a collaboration.
I agree, but I think this is redundant here…
…, which can be leaded back most likely to the different temperament of the hens.
Please cite a paper where it was shown that the examined lines differed in the temperament of the hens. If there is no suitable study, please remove this sentence and also the subsequent outlook on the opportunity of the possibility for the breeding company.
If there is a refence to cite the sentence should be rewritten as follows:
This might give the opportunity for the breeding company to breed and sell calm temperament hybrids, which can be used effectively in those markets where beak trimming is not allowed and laying hens are kept in large group non-caged housing systems.
Round 5
Reviewer 3 Report
Dear authors,
the manuscript was improved according to the comments. I have no further comments.
Regards